# Adsorption of Carbon Dioxide with Ni-MOF-74 and MWCNT Incorporated Poly Acrylonitrile Nanofibers

**DOI:** 10.3390/nano12030412

**Published:** 2022-01-27

**Authors:** Amir Hossein Harandizadeh, Seyedfoad Aghamiri, Mohammad Hojjat, Marziyeh Ranjbar-Mohammadi, Mohammad Reza Talaie

**Affiliations:** 1Department of Chemical Engineering, Faculty of Engineering, University of Isfahan, Isfahan 81746-73441, Iran; harandizadeh.a.h@gmail.com (A.H.H.); m.hojjat@eng.ui.ac.ir (M.H.); 2Department of Textile, Engineering Faculty, University of Bonab, Bonab 55513-95133, Iran; 3Chemical Engineering Department, Faculty of Chemical, Oil and Gas Engineering, Shiraz University, Molasadra St., Shiraz 84334-71946, Iran; mrtalaiekh@shirazu.ac.ir

**Keywords:** CO_2_ adsorption, electrospinning, MWCNT, nanofiber, Ni-MOF-74, secondary growth

## Abstract

Among the new adsorbent forms, nanofiber structures have attracted extra attention because of features such as high surface area, controllable properties, and fast kinetics. The objective of this study is to produce the polyacrylonitrile (PAN) electrospun nanofibers loaded with Ni-MOF-74/MWCNT to obtain maximum CO_2_ adsorption. The prepared PAN/MWCNT/MOF nanofiber based on the Box–Behnken design (BBD) model suggests the CO_2_ adsorption of about 1.68 mmol/g (at 25 °C and 7 bar) includes 14.61 *w*/*v*%, 1.43 *w*/*w*%, and 11.9 *w*/*w*% for PAN, MWCNT, and MOF, respectively. The results showed the effective CO_2_ adsorption of about 1.65 ± 0.03 mmol/g (BET = 65 m^2^/g, pore volume = 0.08 cm^3^/g), which proves the logical outcomes of the chosen model. The prepared PAN/MWCNT/MOF nanofiber was characterized using different analyzes such as SEM, TEM, TG, XRD, FTIR, and N_2_ adsorption–desorption isotherms. More MOF mass loading on the nanofiber surface via secondary growth method resulted in 2.83 mmol/g (BET = 353 m^2^/g, pore volume = 0.22 cm^3^/g, 43% MOF mass loading) and 4.35 mmol/g (BET = 493 m^2^/g, pore volume = 0.27 cm^3^/g, 65% MOF mass loading) CO_2_ adsorption at 7 bar for the first and second growth cycles, respectively. This indicates that secondary growth is more effective in the MOF loading amount and, consequently, adsorption capacity compared to the MOF loading during electrospinning.

## 1. Introduction

Adsorption is an efficient and low energy-consuming process to capture carbon dioxide. [1,2]. Adsorbent structural properties, such as volumetric working capacity, pressure drop, mass transfer, and thermal properties, regulate the performance of adsorptive gas separation processes [3]. These properties are determined by the parameters, such as specific heat, density, the heat of adsorption, porosity, total pore volume, the specific surface area of the adsorbent composite, and also the adsorbent arrangement in adsorption equipment [3,4].

Adsorbents are conventionally used in the form of beads and granules (pellets) in packed bed columns. Consequently, adsorbent particle size can impose disadvantages, such as mass transfer and pressure drop limitations. Therefore, the reduction in particle size is the most common way to decrease mass transfer resistance. However, it can cause more pressure loss [3]. The balance between pressure drop and mass transfer rate usually results in a bed geometry with a large diameter and small height at a particle size around 0.7 mm, which can induce considerable gas maldistribution and channeling. The novel adsorbent structures, such as monoliths, laminates, foams, and fabric structures, can be alternatives to solve the mentioned limitations. Among them, fabric structures due to unique advantages have recently attracted more attention [3,5,6].

The fabric structures are often nonwoven composites that display practical features for gas separation application. These structures are self-supporting and offer high adsorbent loadings, fast heat and mass transfer rates, low attrition rates, high mechanical strength, and low bed pressure drop. Moreover, they can eliminate fluidization problems in the packed bed columns [3,5,6]. Generally, adsorbents embed within the nonwoven structures in the following three ways: (1) development of the physical bonds between adsorbents and fibers, (2) mechanical entrapment of adsorbents within the structure, and (3) a combination of both [5].

The common nonwoven structures, as a substrate for adsorbent loading, are electrospun, meltblown, and spunbond. They show higher filtration efficiencies than conventional adsorbent shaping [5,7,8,9,10]. which is especially true in the case of electrospun nanofibers [11].

Nanofibers are promising adsorbent substrates because of advantages such as high porosity, high surface area, and excellent mechanical properties [12,13,14]. They are produced by applying a high voltage to a polymer solution in the electrospinning process [15,16]. A wide range of materials can be applied to synthesize nanofibers, including natural/synthetic polymers or composites. Besides performing as a supporter, the nanofiber can enhance the adsorption even partially either by intrinsic adsorption [17,18,19] or characteristics induced in nanofiber structure [11,20]. One of the most common polymers in nanofiber fabrication is polyacrylonitrile (PAN) because of its good mechanical properties (bending modulus and failure stress) and continuous formation of nanofibers [21].

Metal-organic frameworks (MOFs) as novel adsorbents have advantages such as high surface area, large pore volume, and a wide variety of structures and compositions [22,23,24]. Millward et al. [25,26] reported that some MOFs have higher saturated CO_2_ capacities compared with traditional zeolites at room temperature. MOFs of the DOBDC (2,5-dioxido-1,4-benzenedicarboxylate) series show different CO_2_ adsorption capacities at the same conditions depending on their metal centers. Therefore, the kind of metal in the center of the DOBDC series is the determining factor in the CO_2_ adsorption capacity [27]. The role of central metal may be related to the different ionic nature of the metal-oxide bonds and unsaturated metal centers (UMCs) in the M/DOBDC series, which are necessary for CO_2_ adsorption at sub-atmospheric pressures [27,28]. Yazaydin et al. [28] investigated experimentally and simulating methods to distinguish MOFs for the highest CO_2_ capacities at a pressure of about 0.1 atm. They considered a lot of MOFs and revealed that Mg/DOBDC and Ni/DOBDC have the highest CO_2_ adsorption capacities at 0.1 atm [28,29]. Their results showed the DOBDC series of MOFs with high ability and open metal sites as the novel candidate for CO_2_ adsorption. Ni/DOBDC has a higher CO_2_ capacity and relatively more stability than Mg/DOBDC at sub-atmospheric pressures [30]. Additionally, Ni/DOBDC has a higher CO_2_ adsorption capacity than commercial adsorbents such as NaX and 5A zeolites at 0.1 atm and 25 °C [31]. In addition, the CO_2_ adsorption capacity of Ni/DOBDC is more stable under the same moist conditions than NaX and 5A zeolites and regenerates at more mild conditions. Therefore, Ni/DOBDC, also known as Ni-MOF-74 or CPO-27-Ni, can be considered as one of the most promising candidates for CO_2_ capture from flue gas [32]. Recently, many investigations have been devoted to new structures of MOFs [21,22,32].

To approach the maximum adsorption in electrospun nanofiber-based adsorbent, loading a sufficient MOF amount on the fibers is needed. However, loading a high amount of MOF causes instability of the electrospinning process [21,33]. Hence, the MOFs secondary solvothermal growth, which is the growth of MOF particles on the nanofibers surface, can be used [21,33,34].

Researchers have also implemented many efforts to improve the nanofibers structure to obtain higher adsorption capacity [22,33,35,36]. Studies have shown that incorporating inorganic compounds in the composition of nanofibers results in a high modulus and high strength structure. Additionally, composite materials with the incorporation of organic–inorganic components have exhibited improved performance [37]. Therefore, the application of carbon nanotubes in the polymer matrixes enhances their mechanical [38] and electrical [39] properties. Moreover, CNT itself is a conventional CO_2_ adsorbent that is located along the axis of nanofiber yarns, and by upgrading the structure properties, it can directly increase the adsorption capacity [11,20]. However, CNTs aggregation occurs due to van der Waals interactions in the polymer solution [40], which causes many voids/defects between agglomerates and the polymer matrix, which affect the mechanical performance of composite fibers. Previous studies show that functionalization, such as carboxylation, can improve the dispersibility of CNTs and reduce the agglomeration [41,42,43,44].

In this study, for the first time in the literature, Ni-MOF-74-incorporated PAN nanofibrous structures were fabricated and used as the CO_2_ adsorbent. In addition, MWCNT was loaded in PAN nanofibers as the adsorbent subsidiary and structure improver. A Box–Behnken design (BBD) of response surface methodology (RSM) was applied to prepare different Ni-MOF-74/MWCNT loaded PAN-based nanofibers to reach the highest CO_2_ adsorption capacities. In addition, to increase the CO_2_ adsorption via the most utilization of MOF crystals in the nanofiber structure, the secondary growth of MOF crystals on nanofibers was inquired. Finally, the fabricated samples were characterized by SEM, TEM, XRD, FTIR, and TG analyses, and N_2_ adsorption–desorption isotherms. After that, the isotherm and CO_2_ adsorption capacity of the MOF/MWCNT incorporated structures were compared with MOF powder.

## 2. Experimental Section

### 2.1. Material

Reagent grade of Nickel acetate and 2,5-dihydroxyterephthalic acid were purchased from Sigma-Aldrich. Dimethylformamide (DMF), Ethanol, and tetrahydrofuran (THF) were purchased from Merck. Polyacrylonitrile (PAN) with a molecular weight of 100,000 (g/mol), was purchased from Polyacryl Company, Isfahan, Iran. MWCNT (Diameter <8 nm, Length ~30 nm, -COOH content ~3.86 wt.%) was provided by Neutrino Company, Tehran, Iran. Deionized (DI) water was used through this work.

### 2.2. MOF Synthesis

The Ni/DOBDC samples were synthesized according to the following hydrothermal method [27,31]. Nickel acetate (0.746 g or 3 mmol) and 2,5-dihydroxyterephthalic acid (0.298 g, 1.5 mmol) were added to 40 mL of 50 vol.% THF in DI water solution. Then the suspension sonicated for half an hour. The mixture was put into the autoclave and heated at 110 °C for 3 days to achieve the yellow crystalline MOF. For activation, the product was rinsed with a mixed solution of methanol, ethanol, and deionized water several times. The resulting MOFs were separated by centrifugation at 6000 rpm. Then MOF particles were dried in an evacuated oven at 100 °C for 2 h and known as Ni-MOF-74.

### 2.3. Electrospinning

To prepare the polymeric solution for electrospinning, the measured amounts of MOF and MWCNT, according to BBD, were mixed with the PAN-DMF solution.

The Polymeric solution injects through a capillary tip with a diameter of 0.7 mm using a 5 mL syringe.

Figure 1 demonstrates the schematic diagram of the electrospinning of PAN/MWCNT/MOF nanofibers. The anode of the high voltage power supply has clamped to the metal collector, and the cathode has been connected to the needle tip.

The applied voltage was 20 KV, the distance between the tip and the collector was 18 cm, and the flow rate of the spinning solution was 0.5 mL/h. The resulting nanofibers accumulated on an aluminum foil-wrapped drum rotating at approximately 250 rpm. Finally, nanofibers were vacuum dried at 100 °C for 12 h.

### 2.4. Design of Experiment/Response Surface Methodology (RSM)

Response surface methodology (RSM) is a suitable method for finding the effects of several independent variables on the response and determining the optimum conditions. Amongst different design of experiment (DOE) methods, many studies have adopted the Box–Behnken design, BBD, because of its need for fewer experiments [45,46].

To determine the effect of three variables, PAN concentration (X1), MWCNT concentration (X2), and MOF concentration (X3) on the CO_2_ adsorption capacity, BBD was used. Each parameter has three levels, which result in 17 experiments, while the CO_2_ adsorption is the response. Table 1 lists variables and levels, which have been chosen through the screening experiments. The standard analysis of variance was employed to analyze the model outputs, and the values of *p* ≤ 0.05 were considered statistically significant [45,46].

### 2.5. Secondary Growth of Ni-MOF-74 on the Nanofiber

The secondary loading of Ni-MOF-74 was performed via hanging up a PAN/MWCNT/MOF nanofiber (Section 2, Section 3 and Section 4) in a Teflon-autoclave during the MOF synthesizing process (as mentioned in Section 2.2). This process was repeated for two consecutive cycles until the nanofiber became saturated. The composite nanofiber was rinsed several times with a mixed solution of methanol, ethanol, and deionized water after each cycle to remove the remaining solvent and activate the adsorbent. Finally, it was placed in an evacuated oven at 100 °C for 2 h.

### 2.6. CO_2_ Adsorption Measurements

An experimental setup based on a volumetric method, Figure 2, was utilized to study the CO_2_ adsorption capacities of Ni-MOF-74 in powder and nanofibers forms. Adsorption capacities are determined by measuring the pressure and temperature of the adsorption cell. The difference between the initial and final pressures of the adsorption cell reveals the equilibrium amount of the adsorbed CO_2_. In a typical test run, 100 mg of a previously degassed sample at 100 °C for 12 h was used. Three similar adsorption/regeneration cycles run for each sample. The regeneration took place in an oven at 100 °C for 4 h. The accuracy in measuring pressure and temperature was 1000 Pa and 0.1 K, respectively.

### 2.7. Characterizations

The crystalline structure of the synthesized adsorbent was evaluated by the XRD method using a D_8_ ADVANCE XRD instrument (Bruker Corporation) in the 2θ range of 1–30° using Co Kα radiation (λ = 1.79 Å). The crystal size of synthesized MOF was calculated by using the Debye Scherrer equation (D = 0.9λ/β cos θ) to the reflections (2θ~7°) [46]. In the mentioned equation, D, λ, β, and θ are the crystal size X-ray wavelength, full width at half-maximum (FWHM) and Bragg angle, respectively. N_2_ adsorption–desorption isotherms of the samples were calculated at 77 K using a Nova instrument (Quantachrome NovaWin2, Boynton Beach, FL, USA) and also a BELSORP MINI instrument (Osaka, Japan). The specific surface area was estimated by the BET method. Field emission-scanning electron microscopy (FESEM, JEOL JSM 6490, JEOL Ltd., Tokyo, Japan) analysis was used to demonstrate the surface morphology of the fabricated samples. Besides, transmission electron microscopy (TEM, Philips CM120, Eindhoven, The Netherlands) analysis was used to prove the MWCNT’s existence along with the nanofibers. TGA test was performed using a Bahr STA-503 instrument at a heating rate of 10 K min^−1^ from 298.2 K to 853.2 K under the argon atmosphere. FTIR measurements were performed using a Jasco-Japan (model 6300, Tokyo, Japan) Fourier transform infrared spectrometer, which was configured for transmittance. Spectra were collected from the mid-infrared region (4000−400 cm^−1^) at a resolution of 4 cm^−1^ with the detector positioned perpendicular to the fiber direction.

## 3. Results and Discussion

### 3.1. MOF Synthesis

Figure 3 shows the SEM of the synthesized MOF crystals at different magnifications. The size of polyhedral crystalline particles is in the range of 6–9 µm.

Figure 4 demonstrates the XRD pattern of Ni-MOF-74 powder [47,48,49]. Two main peaks around 7 and 12 degrees confirm the presence of Ni-MOF-74. The crystal size of MOF by the Scherrer method using the mentioned XRD data is calculated to be about 9 µm [50].

The BET data analysis for synthesized Ni-MOF-74 evaluates the surface specific area of about 788 m^2^ g^−1^, the pore volume of 0.38 cm^3^ g^−1^, and the average pore diameter of 1.92 nm, which are in agreement with previous studies [51,52].

### 3.2. Nanofiber Characterizations

Considering the appearance of the neat PAN, PAN/MWCNT, PAN/MOF, and PAN/MWCNT/MOF nanofiber, it can conclude that with adding of MWCNT and MOF to the PAN solution, the PAN white color changes to dark grey, and yellow, respectively as shown in Figure 5. In comparison, the PAN/MWCNT/MOF structure shows a bright gray color, which is evidence of the following materials’ existence.

As shown in Figure 6a, the electrospun neat PAN nanofibers have a uniform, smooth, and bead-free morphology. The average diameter of the produced nanofibers is 136 ± 3 nm. These structures with a nanosized scale have a higher surface to volume ratio, resulting in more MOF loading sites.

In Figure 6b, the MWCNT loaded on the PAN nanofibers are shown as darker stains along the nanofiber. Figure 6c shows that MOF particles appear as polyhedral crystals, randomly scatter, and trap in nanofibers. Figure 6d exhibits that MOF particles in MOF/MWCNT/PAN nanofibers, besides linking to fibers surface, in some parts, are impregnated along fibers and made a swollen shape along with them. However, it seems that MWCNTs are loaded inside the fibers because of their small sizes, so they could not be seen on nanofibers surfaces except minor parts, which are colligated.

The TG analysis of Ni-MOF-74, PAN nanofiber, and PAN/MWCNT/MOF nanofiber are demonstrated in Figure 7. According to TG of MOF powder, the maximum loss occurs at 345–520 °C which is about 30%. While PAN and PAN/MWCNT/MOF nanofibers have similar two-step loss curves, they occur in the ranges of 125 to 225 °C with 27% weight and 328 to 383 °C about 32%, respectively. Therefore, it seems that MOF is stable at high temperatures up to 345 °C, but loading these particles in nanofibers structure does not help thermal stability, probably because of the lower percentage of MOF relative to PAN. According to the regeneration temperature of Ni-MOF-74 adsorbent at about 100 °C, no decomposition would happen in PAN/MWCNT/MOF structure during the regeneration process.

### 3.3. Box–Behnken Design

Table 2 shows parameter levels for each test run suggested by BBD and experimental responses which are CO_2_ adsorption capacities.

Additionally, BBD suggests Equation (1) as a quadratic model to relate CO_2_ adsorption capacity with model variables.
q_t_ = 1.29 + 0.0587C_MWCNT_ + 0.3250C_MOF_ + 0.0575C_PAN_C_MOF_ − 0.0992C_MOF_^^2^^(1)
where q_t_ is adsorption capacity (mmol/g). This equation indicates that MOF concentration plays the most crucial role in increasing the CO_2_ adsorption capacity as shown in Table 3. In contrary, the effects of PAN concentration and MWCNT content are negligible.

Table 4 shows the analysis of the variance of this model. As it shows, *p*-value ≤ 0.0001 refers to suitable fitness and significance of the chosen model. Additionally, the chosen model suggests the highest possible adsorption capacity can be obtained from a solution of PAN (14.61 *w*/*v*%), MWCNT (1.43 *w*/*w*%), and MOF (11.90 *w*/*w*%), which has a CO_2_ adsorption capacity of 1.68 mmol/g. Confirmatory experiments have been conducted at these conditions three times which showed an adsorption capacity of 1.65 ± 0.03 mmol/g. This is an indication of the accuracy of the proposed model.

Table 4 illustrates the statistical parameters obtained from ANOVA. The least *p*-value belongs to MOF concentration, which leads to the importance of this variable.

Figure 8 presents the effect of each variable on the response. By comparing three bilateral plots, one can conclude that both PAN and MWCNT concentrations individually have a limited impact on CO_2_ adsorption capacity. However, increasing the MOF concentration led to a significant enhancement in the response. The enhancement, as mentioned above, is a consequence of the enormous adsorption capacity of MOF particles. A decrease in CO_2_ adsorption, especially in higher MOF concentrations (more than 6%), is attributed to the electrospinning solution limitation for dispersing all of the MOF completely.

Figure 9 explores the response surface plots and counter plots for investigating the relation of any two variables on the response, while the third variable is in its center. According to Figure 9a, increasing the PAN concentration (more than about 11%) in interaction with MWCNT concentration harms CO_2_ adsorption. Moreover, it shows that in the higher concentration of MWCNT, the lower concentration of PAN is necessary for achieving maximum adsorption. This phenomenon could be related to increasing the solution viscosity induced by MWCNT enhancing. Hence, to adjust the solution viscosity and to obtain the highest adsorption, the PAN concentration should be reduced to get the suitable electrospinning without any MWCNT agglomeration and bead on the nanofiber morphology.

In investigating PAN/MOF plots (Figure 9b), it can be seen that at low concentrations of MOF (less than 6%), increasing the PAN concentration has no significant effect on the CO_2_ adsorption capacity. However, at the higher concentrations of MOF, adsorption capacity intensely depends on PAN concentration and enhances with increasing the PAN concentration. This phenomenon is presumably because of the considerable reduction in the solution viscosity due to the higher MOF concentrations. In a low concentration of MOF, viscosity reduction is less, and MOF particles load on nanofibers, completely. While at higher MOF concentrations solution viscosity intensely drops so that a lot of the MOF particles cannot properly load in nanofibers. Thus, an increase in PAN concentration helps the solution to encompass more MOF crystals and, thus, load more MOF crystals on the nanofiber substrate, which results in higher CO_2_ adsorption capacities.

The MWCNT/MOF plots in Figure 9c show that in low concentrations of MOF, increasing the MWCNT induces a slight increase in CO_2_ adsorption. However, at high concentrations of MOF (more than 6%), increasing the MWCNT concentration have a strong positive impact on the CO_2_ adsorption. It seems that in these concentrations, increasing MWCNT besides its structural improvement has a determining role in compensating solution viscosity, which misses during MOF addition.

### 3.4. Characterization of MOF Secondary Growth on the PAN/MWCNT Nanofiber

According to Figure 10, the first cycle of the secondary growth yields a uniform distribution of MOF crystals on the nanofiber mat. In the second cycle, the MOF crystals fill the remaining spaces on the nanofiber mat. They saturate the nanofiber surface, so the extra MOF crystals deposit on each other and induce the agglomerated MOF particles.

Figure 11 shows the N_2_ adsorption–desorption isotherm of PAN-based electrospun nanofiber loaded with different percentages of MOF/MWCNT and Ni-MOF-74 powder. The uptake of N_2_ molecules at a low relative pressure range (0 < P/Po < 0.1) and moderate pressure (0.5 < P/Po < 0.6), respectively, correspond to the relative volume of the micropores and the existence of mesopores [53]. The insignificant adsorption volume of the pure PAN-based nanofiber indicates the meager existence of pores in the nanofibers. With the MOF addition, there is a noticeable adsorption increase in the low relative pressure zone. A steeper increase happens in samples with higher loading of MOF, which is evidence of the presence of more micropores [54].

The isotherms of the MOF powder and the first and second cycles of MOFs growth resembled type I, according to IUPAC classification, indicating micropore domination in this structure. In contrast, the isotherms of the neat PAN nanofiber and PAN/MWCNT/MOF nanofiber are close to type II, which is characteristic of materials with less porosity or large cavities (macropores) [55].

Table 5 summarizes the corresponding BET surface area and pore structure information of PAN-based electrospun nanofibers loaded with different percentages of MOF/MWCNT and MOF powder. The improvement of the BET surface area and pore volume is evident in the excess MOF amount addition. However, the BET of the second cycle of MOFs growth compared to the first cycle of MOFs growth shows a poor performance despite more MOF loading, which can be a consequence of a blockage of pores due to MOF crystals agglomeration (as shown in Figure 10).

Figure 12 demonstrates the XRD patterns of Ni-MOF-74 powder, PAN/MWCNT, PAN/MOF, and PAN/MWCNT/MOF nanofiber. In the pattern of Ni-MOF-74 powder, two characteristic peaks, narrow and strong, are detectable at 2θ = 7° (110) and 12° (300) [49]. PAN/MWCNT nanofiber represents a broad peak at the range of 2θ = 15–22° (200) and a weak peak at 2θ = 31° (020), which are attributed to the crystal structure of PAN [56]. Furthermore, there are two characteristic peaks of MWCNT, one weak peak is at 2θ = 42° indicating the graphite-like structure of MWCNT and another peak is in the range of 2θ = 28–33°, which seems very difficult, due to the low loading of MWCNT and overlapping with the peak of PAN [57]. As shown in the both XRD pattern of PAN/MOF and PAN/MWCNT/MOF, the mentioned characteristic peaks of Ni-MOF-74 and MWCNT remained constant and in agreement with the structure of Ni-MOF-74. It is important to note that, all samples were successfully prepared without any damage to the crystal structure of the MOF. Such observations have been reported by another researcher [58].

The FTIR spectra of all the samples including Ni-MOF-74 powder, MWCNT, PAN/MWCNT, PAN/MOF, and PAN/MWCNT/MOF nanofibers are shown in Figure 13. In the Ni-MOF-74 spectrum, the characteristic peaks at around 1558, 1419 cm^−1^ are ascribed to stretching vibrations of symmetric and asymmetric carboxylate groups (–COO–) [59]. Moreover, the peaks are located at 920, 850, and 3415 cm^−1^, which are attributed to stretching vibration of Ni–O, C–H, and –OH bonds, respectively [58]. The MWCNT spectrum represents the stretching vibration of C–O, C=O carboxylic acid group, CH_2_, and –OH bonds at 1070, 1729, 2914, and 3450 cm^−1^, respectively [60]. Additionally, Figure 13 depicts the PAN spectrums in presence of Ni-MOF-74 and MWCNT. As shown in the mentioned samples, there are characteristic peaks of PAN at around 3460, 2231, 1459, 1738, 1221, and 1080 cm^−1^, which can be attributed to the stretching vibration of the –OH, C≡N, CH_2_, C=O, C–O bonds from ester groups and the C–O bond from the carboxyl groups, respectively [57]. It is worth mentioning that, when comparing pure PAN nanofiber with MWCNT and Ni-MOF-74 incorporated PAN nanofibers, newly several absorption peaks and changes in the location of characteristic peaks were observed, which is due to the interaction between MWCNT and Ni-MOF-74 with the PAN nanofiber [57,58].

### 3.5. CO_2_ Adsorption Measurements

Figure 14 presents the CO_2_ adsorption capacity at 25 °C under 7 bar pressure for PAN nanofiber, PAN/MWCNT/MOF nanofiber, the first cycle of growth MOF, the second cycle of growth MOF, and MOF powder. The PAN nanofiber displays neglectable CO_2_ uptake, whereas the MOF loaded on PAN nanofibers exhibit the CO_2_ adsorption capacity comparable to that of MOF powder at the same pressure. The CO_2_ adsorption capacity of the first and second cycles of the MOFs growth at 25 °C and 7 bar, respectively are 2.83 and 4.35 mmol/g, which are equivalent to 35% and 53% of the adsorption capacity of the MOF powder (8.1 mmol/g). However, their MOF mass loading in the nanofiber structure is about 43 wt.% and 65 wt.%, respectively (Table 5). So, the capacity adsorption was lower than the excepted one, which can be related to MOFs agglomeration and ultimately pores blockage, as confirmed by the BET data (Table 5). For industrial applications, because of disadvantages such as higher pressure drop and high operation costs, the powder form of synthesized MOF must be shaped into common pellets or loaded in new structures, such as monoliths, foams, and nanofibers. In other similar studies, CO_2_ adsorption capacity of MOF pellet was estimated at about 70% CO_2_ adsorption capacity of powder [61], while as shown in this investigation, the CO_2_ adsorption capacity of the second cycle of the MOFs growth was about 20% less than pellet form.

On the other hand, to achieve a high-performance cyclic adsorption process, the indexes such as recovery, purity, productivity, and total energy of the cyclic adsorption process must be comprehensive considered [62], which are determined by the structural parameters, such as the mass and heat transfer coefficients, mass transfer zone, pressure drop, and adsorption capacity [3]. Therefore, the effect of structure on the following parameters, other than adsorption capacity, should be further studied. Thus, it seems that the nanofiber structure, other than lower adsorption capacity, because of special structural properties compared to the common structure (pellet), can potentially be a promising structure that should be considered further.

The cyclic stability of CO_2_ adsorption capacities for the first and second cycles of MOFs growth and MOF powder is presented in Figure 15. As can be observed, the samples held their adsorption capacity, and no considerable adsorption loss has occurred after twenty cycles.

## 4. Conclusions

PAN electrospun nanofibers loaded with MOF/MWCNT were fabricated and applied as the CO_2_ adsorbent. According to the BBD model, the maximum CO_2_ adsorption was about 1.68 mmol/g, at 25 °C and 7 bar, including 14.61 *w*/*v*%, 1.43 *w*/*w*%, and 11.90 *w*/*w*% for PAN, MWCNT, and MOF, respectively. The CO_2_ adsorption in the experimental setup was about 1.65 mmol/g, which is about 20% of the adsorption capacity of the MOF powder. The secondary growth strategy is applied mainly to increase amount of MOF loading, and consequently increase the adsorption capacity. The results of the first and second cycles of the MOF secondary growth displayed 35% and 53% of the adsorption capacity of the MOF powder, respectively, which in comparison to the common form of adsorbent (pellet) is still 20% less. However, other than the adsorption capacity, there are important structural parameters (the mass and heat transfer coefficients, mass transfer zone, pressure drop, etc.) that affect the cyclic adsorption process and, consequently, final performance. Therefore, besides adsorption capacity, the structural properties play a determining role, and to obtain a suitable adsorbent, a trade-off between both of them should be considered. Hence, because of advantages such as high porosity, high surface area, and excellent mechanical properties, the nanofiber structure can potentially be a promising structure that should be given more consideration.

## Figures and Tables

**Figure 1 nanomaterials-12-00412-f001:**
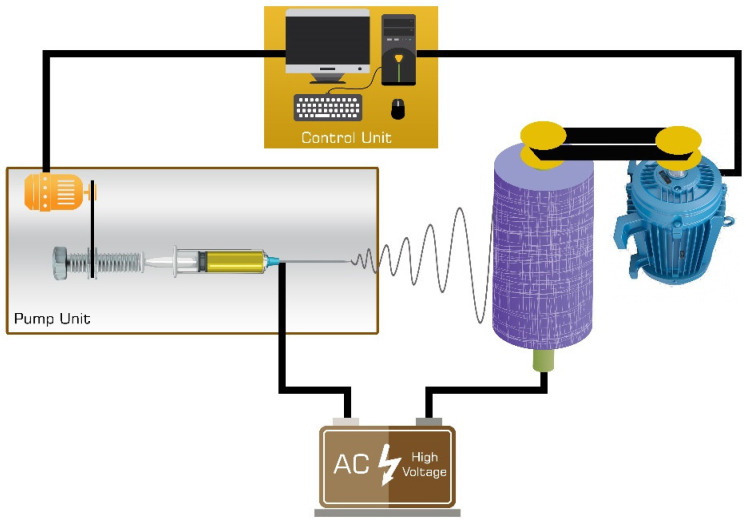
Schematic diagram of electrospinning of PAN/MWCNT/MOF nanofiber.

**Figure 2 nanomaterials-12-00412-f002:**
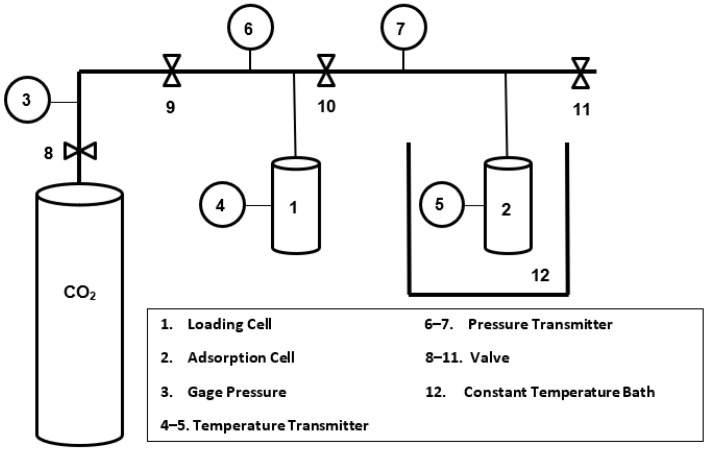
Experimental setup for CO_2_ adsorption evaluation.

**Figure 3 nanomaterials-12-00412-f003:**
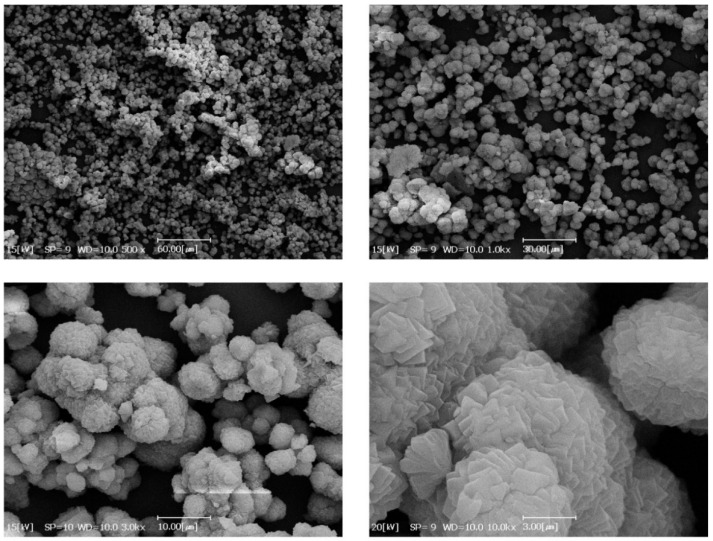
SEM images of synthesized Ni-MOF-74 particles at different magnifications.

**Figure 4 nanomaterials-12-00412-f004:**
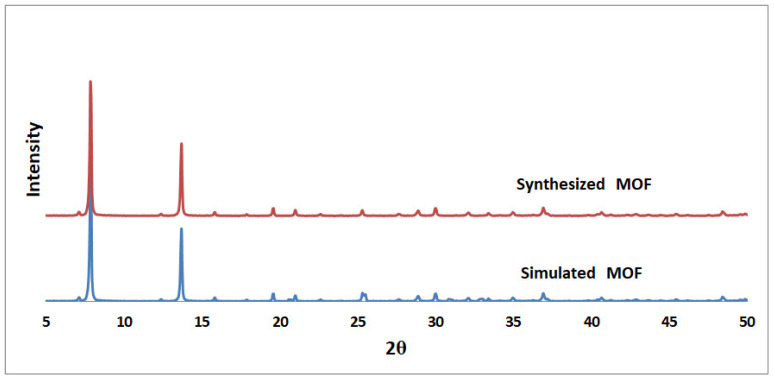
XRD pattern of synthesized Ni-MOF-74 in compared with simulated MOF [48].

**Figure 5 nanomaterials-12-00412-f005:**
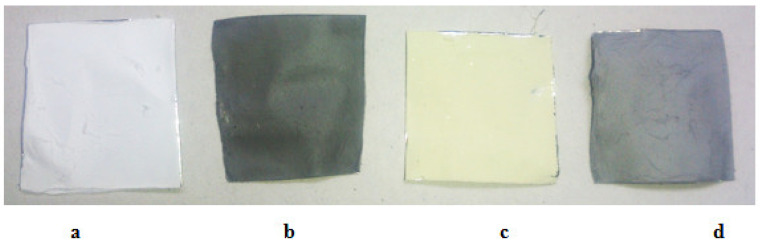
Optical images of electrospun nanofiber mats: (**a**) PAN; (**b**) PAN/MWCNT; (**c**) PAN/MOF; (**d**) PAN/MWCNT/MOF.

**Figure 6 nanomaterials-12-00412-f006:**
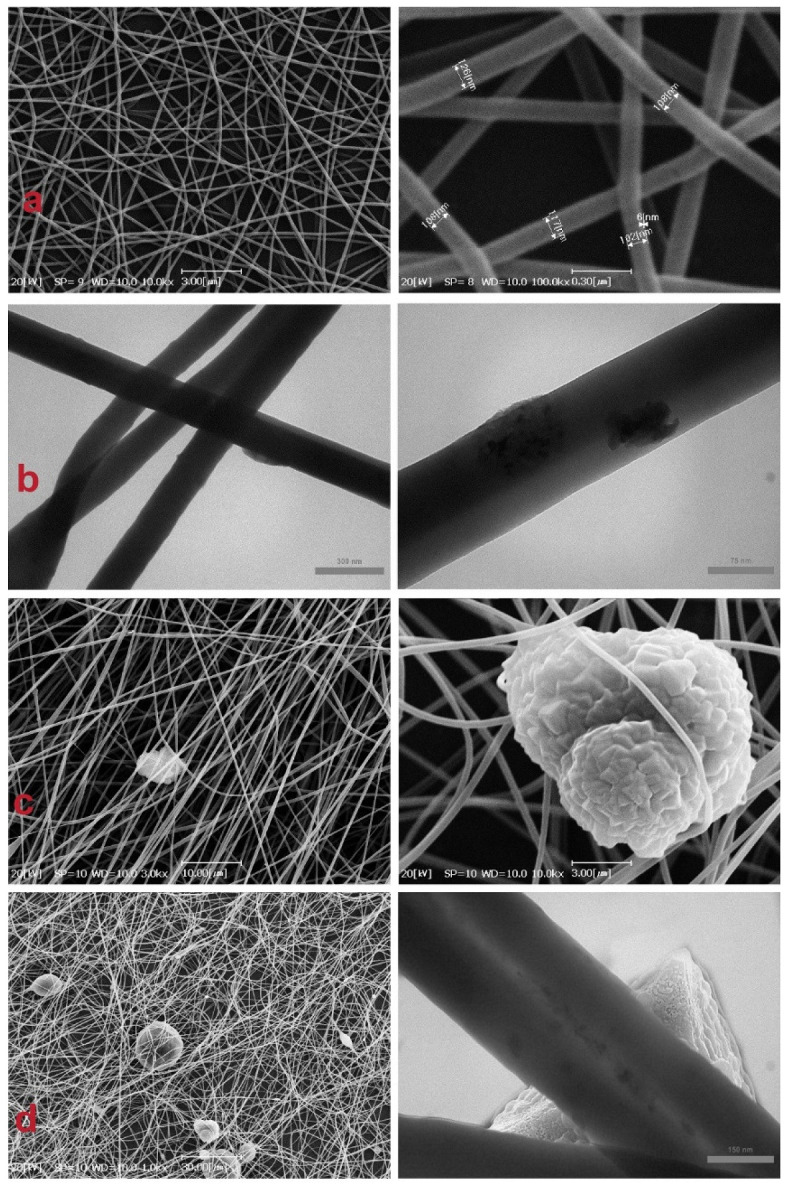
SEM and TEM images of electrospun nanofiber (**a**) PAN; (**b**) PAN/MWCNT; (**c**) PAN/MOF; (**d**) PAN/MWCNT/MOF.

**Figure 7 nanomaterials-12-00412-f007:**
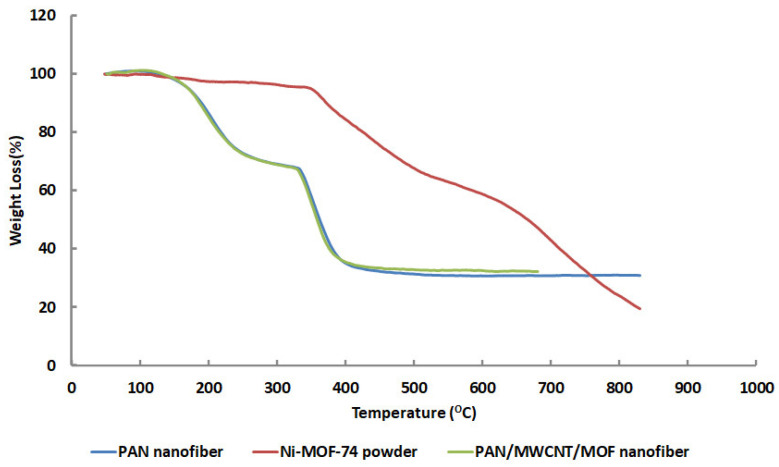
TG analysis of Ni-MOF-74, PAN nanofiber, and PAN/MWCNT/MOF nanofiber.

**Figure 8 nanomaterials-12-00412-f008:**
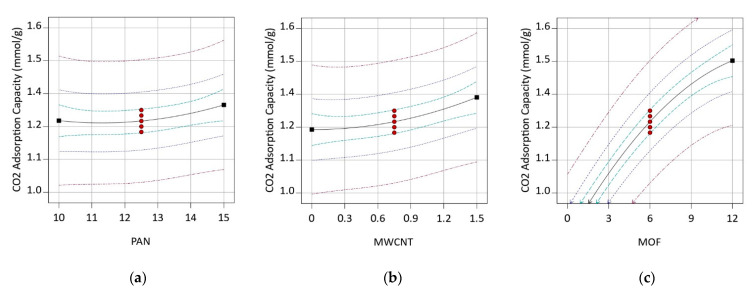
One-factor graphs of variables concentration effect on CO_2_ adsorption, (**a**) PAN (**b**) MWCNT (**c**) MOF.

**Figure 9 nanomaterials-12-00412-f009:**
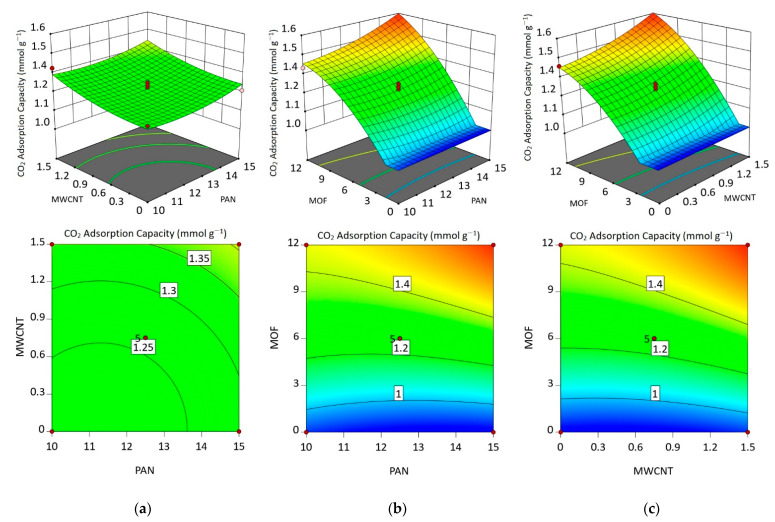
Response surface plots and counter plots on CO_2_ adsorption as follows: effect of (**a**) PAN and MWCNT concentration, (**b**) PAN and MOF concentration, (**c**) MWCNT and MOF concentration.

**Figure 10 nanomaterials-12-00412-f010:**
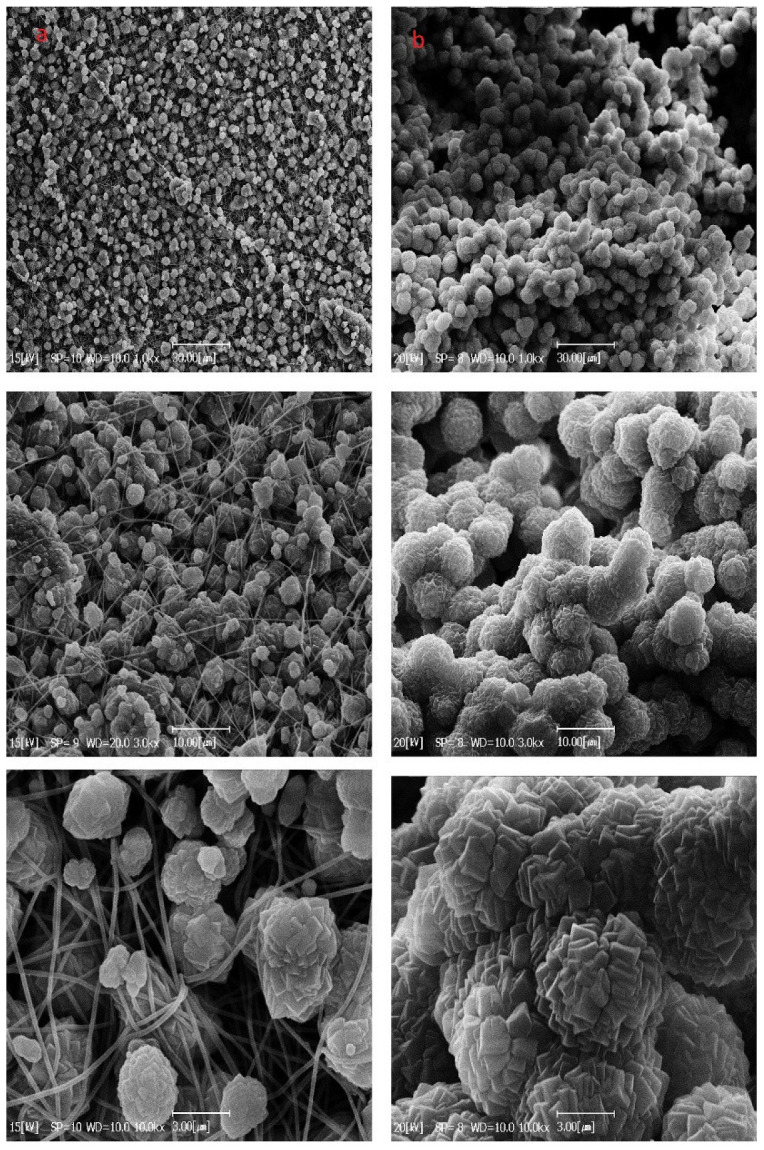
SEM images of MOFs growth on the nanofibers, (**a**) first cycle, (**b**) second cycle, at different magnifications.

**Figure 11 nanomaterials-12-00412-f011:**
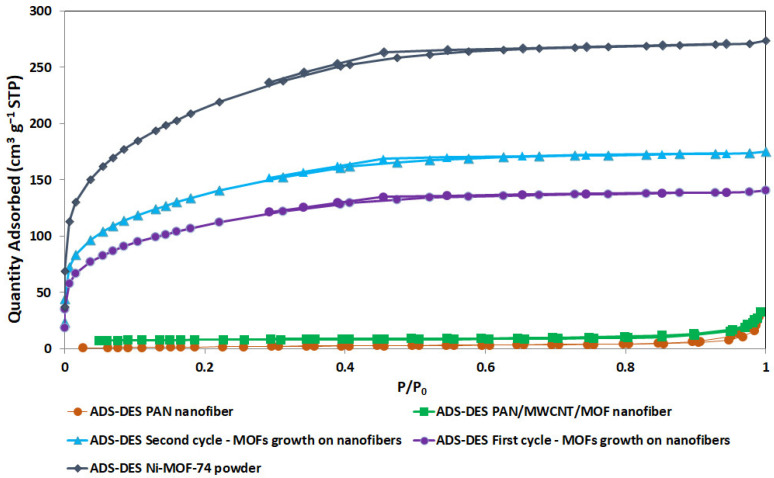
Nitrogen adsorption–desorption isotherms for PAN-based nanofibers loaded with different percentages of MOF/MWCNT and MOF powder.

**Figure 12 nanomaterials-12-00412-f012:**
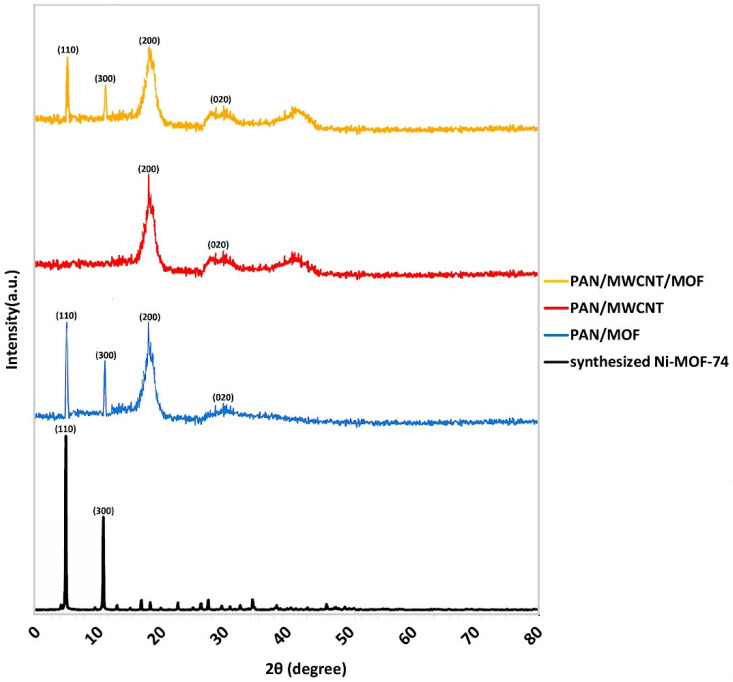
XRD patterns of Ni-MOF-74 powder, PAN/MWCNT, PAN/MOF, and PAN/MWCNT/MOF nanofibers.

**Figure 13 nanomaterials-12-00412-f013:**
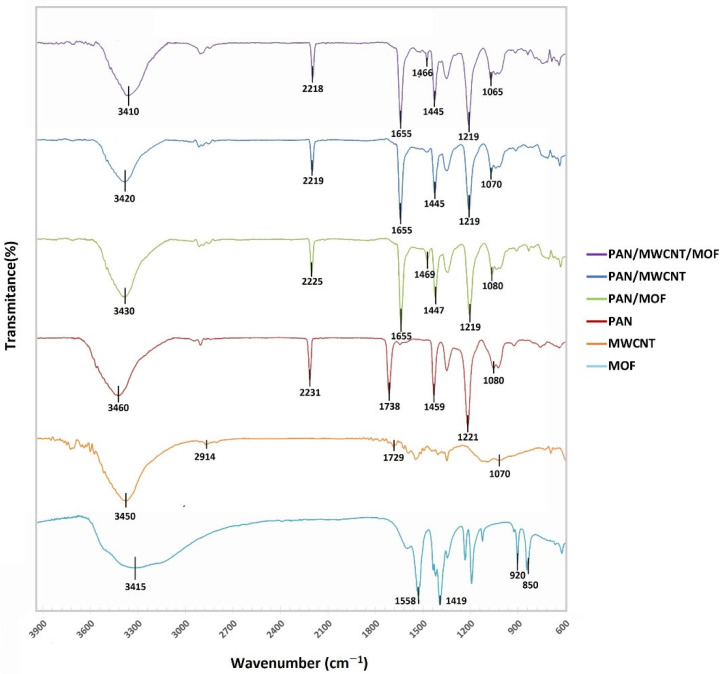
FTIR spectra of Ni-MOF-74 powder, PAN/MWCNT, PAN/MOF, and PAN/MWCNT/MOF nanofibers.

**Figure 14 nanomaterials-12-00412-f014:**
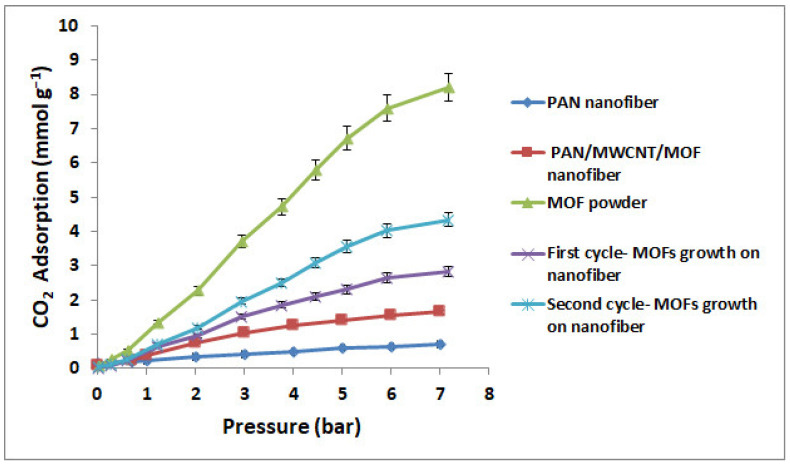
CO_2_ adsorption of PAN nanofiber, PAN/MWCNT/MOF nanofiber, MOF powder, first cycle—MOFs growth on nanofiber, second cycle—MOFs growth on nanofiber, and at 25 °C.

**Figure 15 nanomaterials-12-00412-f015:**
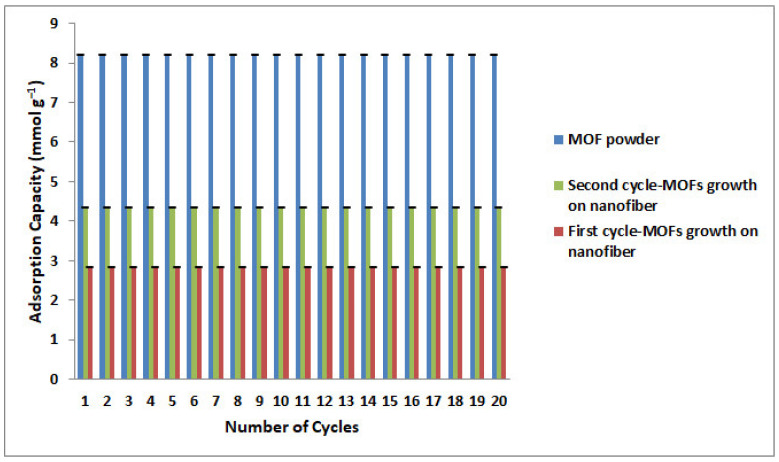
CO_2_ cyclic adsorption capacities of the first cycle—MOFs growth, the second cycle—MOFs growth and MOF powder at 25 °C and 7 bar.

**Table 1 nanomaterials-12-00412-t001:** The variables and levels for Box–Behnken design.

Variable	Name	Level and Quantities		
X1	PAN Concentration	10	12.5	15
X2	MWCNT Concentration	0	0.75	1.5
X3	MOF Concentration	0	6	12

**Table 2 nanomaterials-12-00412-t002:** Experimental design variables and response.

RUN	Variables			Response
	**X1** **(PAN,*****w***/***v*****%)**	**X2** **(MWCNT,*****w***/***w*****%)**	**X3** **(MOF,*****w***/***w*****%)**	**CO_2_ Adsorption** **(mmol CO_2_/g at 25 °C and 7 bar)**
1	15	0	6	1.25 ± 0.03
2	12.5	0.75	6	1.28 ± 0.02
3	12.5	1.5	12	1.59 ± 0.02
4	15	0.75	12	1.63 ± 0.02
5	12.5	0.75	6	1.25 ± 0.02
6	12.5	0.75	6	1.22 ± 0.03
7	10	0	6	0.86 ± 0.03
8	12.5	0.75	6	1.39 ± 0.02
9	15	0.75	0	0.85 ± 0.02
10	10	1.5	6	1.39 ± 0.03
11	12.5	0	0	0.85 ± 0.04
12	10	0.75	0	0.86 ± 0.03
13	12.5	0	12	1.43 ± 0.01
14	12.5	0.75	6	1.26 ± 0.03
15	12.5	1.5	0	0.88 ± 0.02
16	10	0.75	12	1.40 ± 0.03
17	15	1.5	6	1.39 ± 0.03

**Table 3 nanomaterials-12-00412-t003:** Statistical parameters of the ANOVA model.

Source	Sum of Squares	Df	Mean Square	F-Value	*p*-Value	
model	0.9463	9	0.1051	63.04	0.0001≥	significant
A(PAN)	0.0066	1	0.0066	3.96	0.0867	
B (CNT)	0.0276	1	0.0276	16.56	0.0048	
C (MOF)	0.8450	1	0.8450	506.64	0.0001≥	
AB	0.0000	1	0.0000	0.00	1.0000	
AC	0.0132	1	0.0132	7.93	0.0259	
BC	0.0042	1	0.0042	2.53	0.1555	
A^2^	0.0038	1	0.0038	2.27	0.1755	
B^2^	0.0038	1	0.0038	2.27	0.1755	
C^2^	0.0442	1	0.0442	26.52	0.0013	
Residual	0.0117	7	0.0017			
Lack of fit	0.0077	3	0.0026	2.56	0.1931	Not significant
Pure error	0.0040	4	0.0010			
Cor total	0.9580	16				

**Table 4 nanomaterials-12-00412-t004:** Analysis of variance of the quadratic model.

	Sum of Squares	Mean Squares	F-Value	Probe (p) > F	
**Model**	0.9463	0.1051	63.04	0.0001<	significant
**Residual error**	0.0117	0.0017			
**Lack of fit**	0.0077	0.0026	2.56	0.1931	
**Pure error**	0.0040	0.0010			

**Table 5 nanomaterials-12-00412-t005:** Textural properties of neat PAN nanofiber, the first and second cycles of MOFs growth on the nanofibers and the MOF powder.

Samples	Mass Loading of MOF (wt.%)	BET Surface Area (m^2^/g)	Pore Volume (cm^3^/g)	Average Pore Diameter (nm)
PAN nanofiber	0	6.75	0.044	26.1
PAN/MWCNT/MOF nanofiber	12	65	0.08	4.9
First cycle-MOFs growth on nanofibers	43	353	0.22	2.5
Second cycle-MOFs growth on nanofibers	65	493	0.27	2.2
Ni-MOF-74 powder	100	788	0.38	1.9

## Data Availability

The data used to support the findings of this study are included within the article.

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
