# Peer review of "Adsorption of Carbon Dioxide with Ni-MOF-74 and MWCNT Incorporated Poly Acrylonitrile Nanofibers"

_nanomaterials, 2022, doi:10.3390/nano12030412_

Round 1

Reviewer 1 Report

A Ni-MOF-74/MWCNT loaded polyacrylonitrile (PAN) nanofibers with CO2 adsorption capacity was reported. Results showed that the carbon dioxide adsorption capacity of nanofibers after the second growth of MOF (65% MOF loading) was significantly improved. Generally speaking, the manuscript is well written, however, the novelty and importance of this article need to be further clarified. Comments and suggestions are below:

  1. Secondary growth of MOF after electrospinning is a common strategy. Further, the adsorption capacity of the prepared nanofibers material for CO2 is 53% less than that of MOF powder though it could decrease the mass transfer resistance. The application value and innovation of this manuscript should be further considered and a comparison of other works in this area should be provided.

  1. The MWCNT was embedded in the fibers(line 410), in other words, the MWCNT were covered by MOF, which may significantly affect the CO2 adsorption performance? Please provide convincing explanations on this problem.

  1. In Figure 6, only MOF powders on the nanofibers after the second growth can be clearly seen in the SEM images, which is the same as Figure 3. Usually, MOFs particles may fall off from the fibers in the effect of external force, while that cannot be found in the paper, what is the binding force between MOFs and the fibers? Is there any evidence?

Author Response

please see the details in the attachment.

Reviewer 2 Report

The paper presents research on the adsorption of carbon dioxide with Ni-MOF-74 and MWCNT incorporated poly acrylonitrile nanofibers. The presentation of methods and scientific results in the current form is satisfactory for publication in the Nanomaterials journal. The minor and significant drawbacks to be addressed can be specified as follows:
1.    Lines 31 and 32. Secondary Growth ---> Secondary growth.
2.    Line 87. DOBDC? Would you please explain the abbreviation?
3.    Lines 23 and 124. BET is the theory/model, not a method. BET – N2 adsorption (77 K).
4.    For example, lines 149 and 153. PAN/MWCNT/MOF or MOF/MWCNT/PAN? Would you please standardize throughout the paper?
5.    Line 176. 2-2 ---> 2.2.
6.    Fig. 2, Legend. Temperature transmitter ---> Temperature Transmitter.
7.    Fig. 4. Simulated MOF? Please provide the source (the respective refenece(s)) where the data was taken from.
8.    Line between lines 261 and 262. Please add (1) after the equation.
9.    Fig. 11. (i) This figure is illegible. (ii) Only symbols. One color and the same symbols for a given sample (both for the adsorption and desorption curve). (iii) y axis. p/p0 ---> P/P0. See text.
10.    Tab. 5. Mass Loading ---> Mass loading. Average Pore ---> Average pore.
11.    Fig. 12. y-axis. The values are illegible.
12.    Conclusions are too long!!! Please clearly state whether the tested materials (PAN/MWCNT, PAN/MOF, and PAN/MWCNT/MOF) lived up to the hopes placed in them (CO2 adsorption) in comparison with MOF.

Author Response

Review Report 

Comments and Suggestions for Authors

The paper presents research on the adsorption of carbon dioxide with Ni-MOF-74 and MWCNT incorporated poly acrylonitrile nanofibers. The presentation of methods and scientific results in the current form is satisfactory for publication in the Nanomaterials journal. The minor and significant drawbacks to be addressed can be specified as follows:

  1.    Lines 31 and 32. Secondary Growth ---> Secondary growth.

Response: Thank you for your mention, the requested item was corrected.

  1.    Line 87. DOBDC? Would you please explain the abbreviation?

Response: It is a typo and has been modified and highlighted in the revised manuscript.

  1.    Lines 23 and 124. BET is the theory/model, not a method. BET – N2 adsorption (77 K).

Response: The mentioned points were corrected and highlighted in the revised manuscript. The samples were characterized using N2 adsorption-desorption isotherms (77 K). Surface area was calculated according to BET model.

  1.    For example, lines 149 and 153. PAN/MWCNT/MOF or MOF/MWCNT/PAN? Would you please standardize throughout the paper?

Response: Thank you for this point. The standard form is PAN/MWCNT/MOF. It was corrected and highlighted in the whole revised manuscript.

  1.    Line 176. 2-2 ---> 2.2. it is

Response: The mentioned points were corrected and highlighted in the revised manuscript.

  1.    Fig. 2, Legend. Temperature transmitter ---> Temperature Transmitter.

Response: The mentioned point was corrected and highlighted in the revised manuscript.

  1.    Fig. 4. Simulated MOF? Please provide the source (the respective reference(s)) where the data was taken from.

Response: It was added and highlighted in the revised manuscript.

  1.    Line between lines 261 and 262. Please add (1) after the equation.

Response: It was added and highlighted in the revised manuscript.

  1.    Fig. 11. (i) This figure is illegible. (ii) Only symbols. One color and the same symbols for a given sample (both for the adsorption and desorption curve). (iii) y axis. p/p0 ---> P/P0. See text.

Response: The mentioned points were corrected.

  1.    Tab. 5. Mass Loading ---> Mass loading. Average Pore ---> Average pore.

Response: The mentioned points were corrected and highlighted in the revised manuscript.

  1.    Fig. 12. y-axis. The values are illegible.

Response: Thank you, if you mean x-axis was corrected.

  1. Conclusions are too long!!! Please clearly state whether the tested materials (PAN/MWCNT, PAN/MOF, and PAN/MWCNT/MOF) lived up to the hopes placed in them (CO2 adsorption) in comparison with MOF.

Response: Thank you, the conclusion was modified according to your comment. In addition, the following is our main idea regarding this research.

For industrial application, because of disadvantages such as higher pressure drop and high operation costs, the powder form of synthesized MOF must be shaped into common pellets or loaded in new structures such as monoliths, foams and nanofibers, as mentioned in introduction. In other similar studies in this research group, CO2 adsorption capacity of MOF pellet was estimated about 70% CO2 adsorption capacity of powder (Montazerolghaem et al. 2017). While as shown in the present work (Fig. 1, Fig. 13 and Table 5), the CO2 adsorption capacity of second cycle of the MOFs growth was about 4.35 mmol/g which are equivalent to 53% of the adsorption capacity of the MOF powder (8.1 mmol/g) which is about 20% less than pellet form.

On the other hand to achieve a high-performance cyclic adsorption process, the indexes such as recovery, purity, productivity and total energy of the cyclic adsorption process must be comprehensive considered (Haghpanah et al. 2013) which are determined by the structural parameters such as the mass and heat transfer coefficients, mass transfer zone, pressure drop and adsorption capacity (Rezaei 2009). Therefore, the effect of structure on the following parameters other than adsorption capacity should be further studied. So, it seems that the nanofiber structure other than lower adsorption capacity because of special structural properties compared to the common structure (pellet) can be potentially promising structures which should be more considered during the techno-economic study. 

Reference:

F. Rezaei, P. Webley, Optimum structured adsorbents for gas separation processes, Chem. Eng. Sci. 64 (2009) 5182–5191. https://doi.org/10.1016/J.CES.2009.08.029.  M. Montazerolghaem, S.F. Aghamiri, M.R. Talaie, S. Tangestaninejad, A comparative investigation of CO2 adsorption on powder and pellet forms of MIL-101, J. Taiwan Inst. Chem. Eng. 72 (2017) 45–52. https://doi.org/10.1016/j.jtice.2016.12.037. R. Haghpanah, R. Nilam, A. Rajendran, S. Farooq, I.A. Karimi, Cycle synthesis and optimization of a VSA process for postcombustion CO2 capture, AIChE J. 59 (2013) 4735–4748. https://doi.org/10.1002/aic.14192.

Reviewer 3 Report

Hossein et al showing an interesting study about the use of a hybrid material for CO2 adsorption. The gerneral idea is not new, however the used materials are.

Introduction

  • If I am not completely wrong you want to talk about aDsorbents rather than aBsorbents. Please double check in the complete manuscript.
  • Please double check your English and grammar: line 65 They produce à They are produced (fabricated) by …
  • Maybe a problem from my side, but what means DOBDC? Can find the foot note 1.
  • What means characterized by “BET” - Do you you mean characterized by gas adsorption by applying the BET method?

Experimental

  • Please check spelling and grammar!
  • Could you provide a yield for your synthesis?
  • Why wasn’t a commercial instrument used for CO2 adsorption experiments?

Results

  • How was the pore volume and pore diameter calculated? The BET method does not provide such values.
  • How was the pXRD calculated? Why was Co as radiation source used?
  • Re-design figure 7 - add scale ticks
  • Why is the TG signal so noisy?
  • Figure 8 and 9 increase resolution and quality of the figure
  • Figure 12 has a completely different style - please use always the same style for all figures

Conclusion

The general idea is interesting and promising. However, why is not the pure MOF used for CO2 adsorption. Potential disadvanteages are only named, but there is no proof, that the compsitie materials really shows a smaller pressure drop. I am afraid, that the composite materials is still too small and will produce a significant backpressure in a real dynamic adsorption experiment at 7 bar. In addition, at such high pressure a little back pressure does not play a significant role - so I don’t see the advantage of such complicated synthesis routes for producing such a composite material.

Author Response

please see the details in the attachments.

Round 2

Reviewer 1 Report

The paper has been well revised that can be published.

Author Response

Many thanks for the valuable comment of reviewer.

Reviewer 2 Report

The authors have done the essential corrections, provided some detailed answers to some of the questions, and ignored some comments. Overall the manuscript improved. However, one can still find a lot of bugs/errors/typos in the reviewed paper.

1.    Page 2. DOBDC1 ---> DOBDC (2,5-dioxido-1,4-benzenedicarboxylate).

2.    Most of the figures are of very poor quality (they are illegible) - if they are to be published in this quality, it suggests the editor forcing the authors to prepare them in better quality. Most of the figures!!! Especially Figs. 8, 9, and 13.

3.    Fig. 14. Why does x-axis start at -1?

4.    The authors are very inconsistent in notation – see, for example, the legend - Fig. 14. They write Nanofiber once and nanofiber once. I suggest to compare (i) the legends in Figs. 14 and 15 and (ii) the legend in Fig. 14 with the figure captions for this figure. 

5.    There are many such minor errors in notation and the use of upper and lower case letters. I suggest the authors calmly check line by line, paragraph by paragraph, figure by figure, e.g. [35] Electrospinning of Metal – Organic Frameworks for Energy ---> Electrospinning of Metal – organic frameworks for Energy (similarly [2], [9], ... .) [43] ... mystry, "---> mystry",

Author Response

Comments and Suggestions for Authors

The authors have done the essential corrections, provided some detailed answers to some of the questions, and ignored some comments. Overall the manuscript improved. However, one can still find a lot of bugs/errors/typos in the reviewed paper.

  1.    Page 2. DOBDC1 ---> DOBDC (2,5-dioxido-1,4-benzenedicarboxylate).

Response: Thanks, it was corrected.

  1.    Most of the figures are of very poor quality (they are illegible) - if they are to be published in this quality, it suggests the editor forcing the authors to prepare them in better quality. Most of the figures!!! Especially Figs. 8, 9, and 13.

Response: The original of all figures were send. The quality of figures are relative good.

  1.    Fig. 14. Why does x-axis start at -1?

Response: Thanks, it was corrected.

  1.    The authors are very inconsistent in notation – see, for example, the legend - Fig. 14. They write Nanofiber once and nanofiber once. I suggest to compare (i) the legends in Figs. 14 and 15 and (ii) the legend in Fig. 14 with the figure captions for this figure. 

Response: Thanks, there were corrected.

  1.    There are many such minor errors in notation and the use of upper and lower case letters. I suggest the authors calmly check line by line, paragraph by paragraph, figure by figure, e.g. [35] Electrospinning of Metal – Organic Frameworks for Energy ---> Electrospinning of Metal – organic frameworks for Energy (similarly [2], [9], ... .) [43] ... mystry, "---> mystry",

Response: Thanks, there were corrected on the revised manuscrpt.

Reviewer 3 Report

The authors adressed all my comments.

Author Response

Thank you for the valuable comment of reviewer.